

# Detect feature edges for diagnosis of bacterial vaginosis

Jie Li and Yaotang Li

School of Mathematics and Statistics, Yunnan University, Kunming, Yunnan, China

## ABSTRACT

One of the most common diseases among women of reproductive age is bacterial vaginosis (BV). However, the etiology of BV remains unknown. In this study, we modeled the temporal sample of the vaginal microbiome as a network and investigated the relationship between the network edges and BV. Furthermore, we used feature selection algorithms including decision tree (DT) and ReliefF (RF) to select the network feature edges associated with BV and subsequently validated these feature edges through logistic regression (LR) and support vector machine (SVM). The results show that: machine learning can distinguish vaginal community states (BV, ABV, SBV, and HEA) based on a few feature edges; selecting the top five feature edges of importance can achieve the best accuracy for the feature selection and classification model; the feature edges selected by DT outperform those selected by RF in terms of classification algorithm LR and SVM, and LR with DT feature edges is more suitable for diagnosing BV; two feature selection algorithms exhibit differences in the importance of ranking of edges; the feature edges selected by DT and RF cannot construct sub-network associated with BV. In short, the feature edges selected by our method can serve as indicators for personalized diagnosis of BV and aid in the clarification of a more mechanistic interpretation of its etiology.

## INTRODUCTION

Bacterial vaginosis (BV) has been identified to be an independent risk factor for women's health (*Koumans, Kendrick & CDC Bacterial Vaginosis Working Group, 2001*), including preterm delivery, low infant birth weight, development of pelvic inflammatory disease, increased susceptibility to HIV infection, and other chronic health issues (*Hay et al., 1994*; *Ness et al., 2005*; *Sha et al., 2005*; *Atashili et al., 2008*; *van de Wijgert et al., 2008*; *Ma, Forney & Ravel, 2012*). BV is frequently characterized by changes in the vaginal microbiome; however, the causes of these changes are unknown (*Redelinghuys et al., 2020*). Historically, BV has been diagnosed using the Nugent score and/or Amsel's clinical criteria (*Nugent, Krohn & Hillier, 1991*; *Amsel et al., 1983*). The Nugent score is based on the presence or absence of lactobacilli on the Gram stain and generates a score ranging from 1 to 10. A score of ≥7 indicated a positive BV diagnosis. Amsel's criteria focus on clinical symptoms. Three of the following four symptoms yield a positive diagnosis: (1) the presence of a fishy-like odor, (2) the presence of a white discharge, (3) a vaginal pH of >4.5, and (4) the

Corresponding author
Yaotang Li, liyaotang@ynu.edu.cn

detection of a minimum of 20% "clue cells." Amsel's criteria and the Nugent scoring system are considered the "gold standard" for the diagnosis of BV (*Redelinghuys et al., 2020*). However, these methods are difficult to standardize and are subject to interobserver variability because the assessment of diagnostic criteria is dependent on the skill and experience of the observer (*Klebanoff et al., 2004*; *Modak et al., 2011*).

Recent advancements in high-throughput sequencing technologies facilitated the detection of numerous unculturable bacteria from clinical samples (*Adzitey, Huda & Ali, 2013*). Several studies have investigated the relationship between vaginal communities and BV (*Srinivasan et al., 2010*; *Ravel et al., 2011, 2013*; *White et al., 2011*; *Gajer et al., 2012*; *Hickey et al., 2012*; *Ma, Forney & Ravel, 2012*; *Romero et al., 2014*; *Doyle et al., 2018*). For example, *Ravel et al. (2013)* examined the temporal dynamics of 25 vaginal communities over a period of 10 weeks using daily samples collected from healthy women and women diagnosed with symptomatic and asymptomatic BV. *Srinivasan et al. (2010)* conducted deep sequencing of the 16S rRNA gene to investigate the variety and composition of vaginal bacteria in women diagnosed with BV.

Machine learning techniques have been used in this field to diagnose BV (*Baker et al., 2014*; *Beck & Foster, 2014, 2015*; *Pérez-Gómez et al., 2020*; *Loquet et al., 2021*). *Baker et al. (2014)* used numerous feature selection and classification algorithms to uncover the most important features for BV diagnosis. Beck & Foster (2014) utilized three different algorithms—*i.e.*, genetic programming, logistic regression (LR), and random forest—to identify potential diagnostic features and used them for BV diagnosis. However, the diagnostic features selected by the three algorithms were considerably dissimilar. *Beck & Foster (2015)* subsequently selected diagnostic features according to their importance in each classification model and identified largely similar important features from different classification models. *Pérez-Gómez et al. (2020)* used decision tree (DT) and ReliefF (RF) algorithms as feature selectors with classifier support vector machine (SVM) and LR; they then compared their results with those of *Beck & Foster (2015)*. *Loquet et al. (2021)* designed classification and regression trees to diagnose BV in pregnant women.

Existing research indicates that BV is a systemic abnormality caused by multiple bacteria and that interactions between bacteria also play a role in the onset of BV (*Srinivasan et al., 2010*; *White et al., 2011*; *Ravel et al., 2011, 2013*; *Gajer et al., 2012*; *Romero et al., 2014*; *Doyle et al., 2018*). Therefore, studying bacterial interactions is necessary to gain insight into the pathogenesis of BV. Methodologically, the bacterium can be defined as a network node, and interbacterial interactions can be defined as network edges. The current challenge was to identify network feature edges that can characterize the state of the vaginal community. Efforts to find reliable feature edges rely on information related to bacterial interactions; thus, temporal sample datasets are required. The dataset reported by *Ravel et al. (2013)* provides ideal material to investigate this topic. In this paper, we model each temporal dataset of the vaginal community from *Ravel et al. (2013)* to a network and then create 25 networks. We apply supervised feature selection methods to 25 networks to find feature edges that are related to BV, and validate those feature edges using two classification algorithms. To our knowledge, we first used supervised feature selection algorithms to find network feature edges associated with BV.

We hope that these feature edges will aid in the diagnosis of BV and promote research into the pathogenesis of BV.

## MATERIALS AND METHODS

### Vaginal microbiome datasets

The dataset (http://www.ncbi.nlm.nih.gov/bioproject/?term=PRJNA208535) was originally reported by *Ravel et al. (2013)*. *Ravel et al. (2013)* sequenced vaginal communities collected daily for ten weeks from 25 women diagnosed with symptomatic BV (SBV: $n$ = 15 women), asymptomatic BV (ABV: $n$ = 6), or healthy (HEA: $n$ = 4). In total, *Ravel et al. (2013)* sequenced 1,657 samples (median = 67 per woman) and obtained 420 8,757,681 high-quality sequenced reads of the V1–V3 hypervariable region of 16S-rRNA genes, with a median of 5,093 reads per sample. These datasets had been generated >10 years ago, they were reanalyzed using the amplicon sequence variant (ASV) method. The softwares Mothur (version 1.39.5), VSEARCH (version 2.3.4) and USEARCH (version 10.0.240) were used for data analysis. First, we used the command "pcr.seqs" in Mothur to remove the primers of the sequence. We then used VSEARCH to remove sequence redundancy and set the parameter minuniquesize of VSEARCH to 1. We used USEARCH to reduce the noise of the clean sequence to generate a representative ASV sequence and set the parameter minisize of USEARCH to the default value of 8. We used VSEARCH again to produce the ASV table and set the matching ratio to 1 (*i.e.*, complete matching). Finally, we used blastn combined with EzBioCloud database to annotate the obtained ASV. Although ASV can be annotated using the Silva database, the 16S sequence of EzBioCloud database has undergone several manual corrections, which is more accurate than the annotation in the Silva database.

### Feature selection algorithms

Feature selection aims to find the optimal subset of features. Feature selection can be used to eliminate irrelevant or redundant features, reduce the number of features, filter out features related to class information, and improve model accuracy. The general process of feature selection:

- Generate subsets: search for feature subsets and provide feature subsets for the evaluation function;
- Evaluation function: evaluate the quality of the feature subset;
- Stopping criteria: related to the evaluation function, generally a threshold; the search can be stopped after the evaluation function reaches a certain standard;
- Verification process: verify the validity of the selected feature subset on the verification dataset.

DT (*Bramer, 2007*) and RF (*Robnik-Šikonja & Kononenko, 2003*) feature selection algorithms, which belong to the supervised feature selection method, were used in the present study. These methods are implemented on a function-by-function basis in the Python modules skfeature (*Li et al., 2018*) and sklearn.

## Classification algorithms

A classification algorithm has two phases: learning and classification. The classification model is trained on the given dataset and its label information during the learning phase; during the classification phase, the classification model assigns the label to the new dataset. The classification model in this paper uses LR (*Han, Pei & Kamber, 2011*) and SVM (*Wang et al., 2018*), both of which are classic binary classification models that are widely used in a variety of fields.

## Leave-one-out validation

The dataset was divided into the training and validation sets. The training set was used to train the model, whereas the validation set was used to assess the model's generalizability. If the size of the dataset $D$ was $N$, then $N - 1$ pieces of data should be used for training, and the remaining data should be used for validation. A total of $N$ times are calculated for each group taken from D as the verification set until all samples have been verified as the set. Finally, the mean value of the verification error was calculated. This method is called leave-one-out cross-validation (*Torgo, 2010*).

## Performance measures

The classification accuracy for each model was measured using the accuracy ratio (=(TP + TN)/(P + N)), precision ratio (=(TP)/(TP + FP)), and recall ratio (=TP/P), where P, N, TP, TN, FP, and FN are the positive, negative, true positive, true negative, false positive, and false negative prediction values in the confusion matrix, respectively.

## Experimental studies

We obtained the ASV time series data of 25 vaginal communities. The ASV is regarded as the node of the network. The interaction between ASVs is used to construct the network edge, and the weight of the network edge was determined through the correlation coefficient between ASVs. Therefore, an ASV correlation coefficient network was constructed, which is an undirected and weighted network, and different labels (ABV, SBV, HEA) were assigned to the network according to the diagnosis of vaginal community. The correlation coefficient was calculated using the "spacc" function in the R package SpiecEasi, and the parameters in the "spacc" function were set as default values. Because there are too many zeros in the time series of some ASVs, in the process, we summed the counts of ASVs across the 25 datasets and selected ASVs with sum of counts in the top 60 to construct the networks. In 25 datasets, the counts sum of top 60 ASVs has proportion of 0.996 in the sum of all ASVs counts (Fig. S1). This ensures that the time series of ASVs contain many zero values cannot be selected. It is statistically meaningless to use them to construct network. The entire calculation process was performed using R-script (ASVCN. R). The weight of the edge of the network was extracted, and a vector (called network feature vector) was constructed to characterize the corresponding network. In total, we obtained 25 network feature vectors. Feature selection algorithms (DT and RF) were subsequently used to explore the network feature edge (python script: FeatureSelection.py) that plays an important role in the classification of vaginal community status.

In the present study, we aimed to determine the network feature edges that can distinguish SBV from ABV, ABV from HEA, SBV from HEA, and BV (SBV+ABV) from HEA. Therefore, the network feature vectors were categorized into four corresponding groups of data, and the grouped network labels were digitized. For example, the dataset of SBV *vs* ABV group only contained network feature vectors labeled SBV and ABV, and maps the SBV label to 1 and the ABV label to 0. In each grouping, owing to the small number of network feature vectors (SBV *vs* ABV: 21; ABV *vs* HEA:10; SBV *vs* HEA:19; BV *vs* HEA:25), we used the Leave-One-Out method to score the importance of network edges. For example, for grouping SBV *vs* ABV, each time one network feature vector was left, others 20 feature vectors were used as predictors to input the feature selection algorithm (DT and RF) which can generate the importance values of network edges. We repeated this process 21 times, ensured that each feature vector was left once and only once, then the whole process scored each network edge 21 times. Finally, the average of these scores was calculated as a measure of the importance of network edges, and the network edge was sorted according to size to facilitate the selection of network feature edges.

To verify whether the selected network feature edges can distinguish the community status and potentially be used as a marker for BV diagnosis, we selected the network feature edge as the predictors and verified it with the classification algorithm (SVM and LR). Specifically, according to the ranking results of the importance of network edges, the top k (=5, 10, 15, 20, and 25) network feature edges with high scores were selected as the predictors of the classification algorithm. Moreover, the corresponding label vector was used as the response vector of the classification algorithm to input the classification algorithm. During this process, we also used the Leave-One-Out method to randomly select a network feature vector as the test set and other network feature vectors as the training set. After the classifier training, we predicted the label of the test set. After implementing the Leave-One-Out method, the classification evaluation index was calculated according to the predicted value and actual value of the label of the network feature vector. We implemented the Leave-One-Out method 20 times and finally calculated the classification evaluation indictors, which reflects the overall ability of the selected network features to identify the states of the community.

## RESULTS

In results, we show the feature edges obtained using the RF and DT algorithms as feature selectors and their mean importance value (MIV). The MIV was the mean importance value obtained by each feature across all runs of the feature selection method. The feature edges in the tables were ranked by MIV.

In addition, we show the performance of the classification algorithm with feature selection methods. The performance was measured using Accuracy (ACC), Precision (Pre), and Recall (Recall). As described in Experimental Studies, 20 times of the LR/SVM were performed with different feature edges, and the values of ACC, Pre, and Recall in the tables are the mean values across 20 times.

**Table 1  Top five feature edges obtained using the feature selection algorithms for BV *vs* HEA group.**

| DT | | RF | |
|---|---|---|---|
| **Feature edges** | **MIV** | **Feature edges** | **MIV** |
| Megasphaera.AFUG_s Prevotella.amnii | 0.12 | Mobiluncus.mulieris Peptoniphilus.coxii | 1,746.08 |
| Parvimonas.KQ959647_s Anaerococcus.tetradius | 0.11 | Moryella.AY995258_s Mobiluncus.mulieris | 1,734.28 |
| Lactobacillus.crispatus Peptostreptococcus.anaerobius | 0.09 | Moryella.AY995258_s Peptoniphilus.coxii | 1,712.32 |
| AF125206_g.DQ666092_s Dialister.propionicifaciens | 0.09 | Moryella.AY995258_s Peptoniphilus.lacrimalis | 1,688.60 |
| Dialister.KQ960846_s Haemophilus.JH591066_s | 0.08 | Streptococcus.oralis Peptoniphilus.lacrimalis | 1,656.40 |

**Table 2  Performance measures obtained using the classifiers in experiment BV *vs* HEA.**

DT/RF

| | LR | | | SVM | | |
|---|---|---|---|---|---|---|
| **Feature number** | **Acc** | **Pre** | **Recall** | **Acc** | **Pre** | **Recall** |
| 5 | 1.0/0.64 | 1.0/0.83 | 1.0/0.71 | 0.92/0.84 | 0.91/0.84 | 1.0/1.0 |
| 10 | 1.0/0.52 | 1.0/0.85 | 1.0/0.52 | 0.84/0.84 | 0.84/0.84 | 1.0/1.0 |
| 15 | 0.8/0.52 | 1.0/0.80 | 0.76/0.57 | 0.84/0.84 | 0.84/0.84 | 1.0/1.0 |
| 20 | 0.8/0.68 | 1.0/0.84 | 0.76/0.76 | 0.84/0.84 | 0.84/0.84 | 1.0/1.0 |
| 25 | 0.8/0.60 | 1.0/0.82 | 0.76/0.67 | 0.84/0.84 | 0.84/0.84 | 1.0/1.0 |
| Mean | 0.88/0.60 | 1.0/0.83 | 0.86/0.65 | 0.85/0.84 | 0.85/0.84 | 1.0/1.0 |

## BV *vs* HEA results

In Table 1, the feature selection algorithms investigated in this work obtained different results. They shared no common feature edges in their top 5 rankings.

Table 2 shows that LR performs better in DT feature edges than SVM; conversely, SVM performs better in RF feature edges than LR; the measures (ACC, Pre, and Recall) of LR with DT feature edges can achieve 1 in the top 5 and 10 feature edges.

## SBV *vs* HEA results

Table 3 shows that feature selection algorithms also obtained different results from each other and did not share common feature edges in the top 5 rankings. The DT results show that the MIV (0.42) of the feature edge Lactobacillus.jensenii-Peptoniphilus.KQ960236_s is higher than that of other feature edges.

Table 4 shows that the performance of LR and SVM with DT results is better than that with RF results. Both the measures of LR and SVM achieve 1 in the top 5 feature edges of DT results. The measures of SVM also achieve 1 in the top 15 feature edges of DT results.

**Table 3 Top five feature edges of importance for SBV *vs* HEA group selection by DT/RF feature selection algorithms.**

| DT | | RF | |
|---|---|---|---|
| **Feature edges** | **MIV** | **Feature edges** | **MIV** |
| Lactobacillus.jensenii Peptoniphilus.KQ960236_s | 0.42 | Mobiluncus.mulieris Peptoniphilus.coxii | 1,713.63 |
| Atopobium.vaginae Atopobium.AEDQ_s | 0.05 | PAC002181_g.JN713504_s Streptococcus.oralis | 1,664.95 |
| Megasphaera.ADGP_s Streptococcus.agalactiae | 0.05 | Peptoniphilus.lacrimalis Peptoniphilus.coxii | 1,661.32 |
| Megasphaera.AFUG_s Prevotella.amnii | 0.05 | Moryella.AY995258_s Peptoniphilus.lacrimalis | 1,660.11 |
| Mageeibacillus.indolicus Peptococcus.niger | 0.05 | Moryella.AY995258_s Mobiluncus.mulieris | 1,644.74 |

**Table 4 Performance measures obtained using the classifiers in experiment SBV *vs* HEA.**

| DT/RF | | | | | | |
|---|---|---|---|---|---|---|
| | LR | | | SVM | | |
| **Feature number** | **Acc** | **Pre** | **Recall** | **Acc** | **Pre** | **Recall** |
| 5 | 1.0/0.58 | 1.0/0.82 | 1.0/0.60 | 1.0/0.79 | 1.0/0.79 | 1.0/1.0 |
| 10 | 0.95/0.53 | 1.0/0.75 | 0.93/0.60 | 0.95/0.79 | 1.0/0.79 | 0.93/1.0 |
| 15 | 0.95/0.63 | 1.0/0.83 | 0.93/0.67 | 1.0/0.79 | 1.0/0.79 | 1.0/1.0 |
| 20 | 0.95/0.63 | 1.0/0.90 | 0.93/0.60 | 0.79/0.79 | 0.79/0.79 | 1.0/1.0 |
| 25 | 0.95/0.53 | 1.0/0.80 | 0.93/0.53 | 0.79/0.79 | 0.79/0.79 | 1.0/1.0 |
| Mean | 0.96/0.58 | 1.0/0.82 | 0.94/0.60 | 0.91/0.79 | 0.92/0.79 | 0.99/1.0 |

## ABV *vs* HEA results

Table 5 shows that the feature selection algorithms also obtained different results from each other. Although there are no common feature edges in the two results, they have a common node (Lactobacillus.iners).

Table 6 shows that the measures of LR with DT results achieve 1 in the top five feature edges. The measures of SVM with DT results achieve 1 in the top 10 feature edges.

## SBV *vs* ABV results

Table 7 shows that the feature edges in two results are different. The node (Lactobacillus.iners) appears in two results. The MIV of edge (JH591066_s-Gemella.haemolysans) is bigger than that of edges.

Table 8 shows that no measures of LR/DT, LR/RF, SVM/DT, and SVM/RF achieve 1. However, SVM/DT has better performance in the top 15 feature edges. The performances of the feature edges selected by the feature selection methods are not satisfactory.

Based on the abovementioned results, the following conclusions can be inferred:

**Table 5 Top five feature edges of importance for ABV *vs* HEA group selection by DT/RF feature selection algorithms.**

| DT | | RF | |
|---|---|---|---|
| **Feature edges** | **MIV** | **Feature edges** | **MIV** |
| Lactobacillus.iners Lactobacillus.jensenii | 0.1 | Lactobacillus.iners Lactobacillus.crispatus | 1,527.4 |
| Lactobacillus.iners Peptoniphilus.KQ960236_s | 0.1 | Lactobacillus.iners Lactobacillus.fornicalis | 1,462.5 |
| Gardnerella.vaginalis Shuttleworthia.AY959069_s | 0.1 | Atopobium.vaginae PAC002181_g.JN713504_s | 1,432.2 |
| Streptococcus.JWEZ_s Peptostreptococcus.anaerobius | 0.1 | Lactobacillus.iners Streptococcus.agalactiae | 1,399.6 |
| Dialister.KQ960846_s Haemophilus.JH591066_s | 0.1 | Lactobacillus.crispatus Gardnerella.CP019058_s | 1,361.5 |

**Table 6 Performance measures obtained using the classifiers in experiment ABV *vs* HEA.**

DT/RF

| | LR | | | SVM | | |
|---|---|---|---|---|---|---|
| **Feature number** | **Acc** | **Pre** | **Recall** | **Acc** | **Pre** | **Recall** |
| 5 | 1.0/0.6 | 1.0/1.0 | 1.0/0.33 | 0.8/0.20 | 0.75/0.33 | 1.0/0.33 |
| 10 | 0.7/0.6 | 1.0/1.0 | 0.5/0.33 | 1.0/0.30 | 1.0/0.43 | 1.0/0.50 |
| 15 | 0.7/0.5 | 1.0/0.67 | 0.5/0.33 | 0.9/0.30 | 1.0/0.43 | 0.83/0.50 |
| 20 | 0.7/0.8 | 1.0/1.0 | 0.5/0.67 | 0.8/0.40 | 0.83/0.50 | 0.83/0.67 |
| 25 | 0.7/0.8 | 1.0/1.0 | 0.5/0.67 | 0.7/0.40 | 0.80/0.50 | 0.67/0.67 |
| Mean | 0.76/0.66 | 1.0/0.93 | 0.60/0.47 | 0.84/0.32 | 0.88/0.44 | 0.87/0.53 |

**Table 7 Top five feature edges of importance for SBV *vs* ABV group selection by DT/RF feature selection algorithms.**

| DT | | RF | |
|---|---|---|---|
| **Feature edges** | **MIV** | **Feature edges** | **MIV** |
| JH591066_s Gemella.haemolysans | 0.60 | Peptoniphilus.lacrimalis Peptoniphilus.coxii | 1,612.38 |
| JWEZ_s KQ959671_s | 0.12 | Streptococcus.oralis Peptoniphilus.lacrimalis | 1,583.24 |
| Lactobacillus.jensenii KQ960236_s | 0.05 | Lactobacillus.iners Lactobacillus.crispatus | 1,563.19 |
| Staphylococcus.lugdunensis KQ960846_s | 0.04 | Mobiluncus.mulieris Peptoniphilus.coxii | 1,536.43 |
| Lactobacillus.iners Proteus.mirabilis | 0.01 | JYGP_s Peptoniphilus.coxii | 1,527.43 |

**Table 8 Performance measures obtained using the classifiers in experiment SBV *vs* ABV.**

DT/RF

| Feature number | LR | | | SVM | | |
|---|---|---|---|---|---|---|
| | Acc | Pre | Recall | Acc | Pre | Recall |
| 5 | 0.81/0.57 | 1.0/0.71 | 0.73/0.67 | 0.86/0.71 | 0.83/0.71 | 1.0/1.0 |
| 10 | 0.86/0.52 | 1.0/0.78 | 0.8/0.47 | 0.81/0.71 | 0.79/0.71 | 1.0/1.0 |
| 15 | 0.86/0.48 | 1.0/0.70 | 0.8/0.47 | 0.90/0.71 | 0.88/0.71 | 1.0/1.0 |
| 20 | 0.86/0.52 | 1.0/0.78 | 0.8/0.47 | 0.76/0.71 | 0.75/0.71 | 1.0/1.0 |
| 25 | 0.86/0.57 | 1.0/0.80 | 0.8/0.53 | 0.71/0.71 | 0.71/0.71 | 1.0/1.0 |
| Mean | 0.85/0.53 | 1.0/0.75 | 0.79/0.52 | 0.81/0.71 | 0.79/0.71 | 1.0/1.0 |

- Machine learning can distinguish vaginal community states (BV, ABV, SBV, and HEA) based on a few feature edges (bacterial interaction). The classification performance between BV, SBV, ABV, and H is better than that between SBV and ABV.
- Selecting the top five feature edges of importance can achieve the best accuracy for the feature selection and classification model. In some cases, the more feature edges will decrease the performance of the classification algorithm.
- The feature edges selected by DT outperform those selected by RF in terms of classification algorithm LR and SVM. However, LR with DT feature edges is more suitable for diagnosing BV.
- The results of two feature selection algorithms exhibit differences in the importance of ranking of edges. Taking the top five feature edges as an example, the feature edges chosen by the two algorithms have almost no intersection.

## DISCUSSION

The feature edges identified in the present study can be used to distinguish the state of the vaginal microbiome (BV *vs* HEA; SBV *vs* HEA; ABV *vs* HEA); however, the ability to distinguish between SBV and ABV is limited. In conclusion, our results show that there are differences in the expression of feature edges (interaction between the bacteria) under different vaginal environmental conditions. Naturally, these feature edges may be useful in diagnosing BV. The feature edges chosen by different feature selection algorithms are inconsistent, the same problem that has also been observed in previous studies (*Baker et al., 2014*). This adds to the complexity of the interpretability of feature edges. Similarly, *Ma & Ellison (2021)* found 15 different types of network markers (motif, interactions among three species) that were present only in the BV microbiome and absent in the healthy microbiome, which were validated on other BV datasets. Compared with the results of *Ma & Ellison (2021)*, we found almost no overlap with our results. This implies that the identification of BV associate feature edges may not be unique and that finding universal feature edges is difficult and complex, necessitating the mining of more sample data.

The BV "single causative agent" theory is no longer widely accepted. Alternatively, BV is thought to be polymicrobial in nature. However, the pathogenesis of BV is still poorly understood. There is evidence showing that interspecies interactions characterize the vaginal microbiota with BV. Gardnerella spp. may provide a favorable environment for the growth of other BV-associated bacteria during the onset of BV, according to *Pybus & Onderdonk (1997)*. *Srinivasan & Fredricks (2008)* proposed that BV occurs when BV-associated bacteria enter the vagina and displace lactobacilli. Furthermore, BV-associated bacteria (Bacteroides spp., *Enterococcus faecalis*, Vaginal G., Mobiluncus spp., and Peptoclococcus spp.) can inhibit Lactobacillus growth. And in a healthy vaginal environment, lactobacillus species produce hydrogen peroxide ($H_2O_2$) to inhibit the overgrowth of anaerobic bacteria. The reduction of Lactobacillus spp. was therefore considered to indicate vaginal dysbiosis. Those arguments imply, logically, that interactions between certain bacteria are related to BV. Identifying the critical pathway of interactions can shed insights into the ecological mechanisms of BV. Feature edges (interaction between bacteria) have the potential to reveal the dysbiosis pathway and signaling associated with BV. Further, with the modelling vaginal community as a complex ecosystem, it has also been hypothesized that BV corresponds to one or more non-equilibrium states in the complex ecosystem (*Fredricks, Fiedler & Marrazzo, 2005*; *Ravel et al., 2011*; *Ma, Forney & Ravel, 2012*). Detecting certain 'signatures' of BV states is reducible to finding a sub-network (*i.e.*, network motif). The network feature edges selected by our method should belong to the sub-networks. However, surveying our results, the network feature edges cannot construct sub-network that we desired (Tables 1, 3, 5 and 7). This implies that the detecting of the sub-networks associated BV is more complex and our method has limitation on this work. In addition, the pathogenesis of BV cannot be completely determined by bacteria. Several risk factors have been identified in the pathogenesis of BV, such as age, socio-economic status, antibiotic usage, sexual behavior, and ethnicity (*Brumley, 2012*; *Singh Amita & Sumitra Nain, 2015*; *Ranjit et al., 2018*), thereby indicating that the pathogenesis of BV is complex. The road to discovering the full pathogenesis of BV remains long. Our research provides important candidate materials (feature edges) and tools to further our understanding of BV risk and etiology.

## CONCLUSION

The feature edges discovered by the feature selection algorithm can accurately distinguish BV and the health status of the vaginal microbiome. These features can also help reveal the pathogenesis of BV. The feature edges selected by different feature selection algorithms are varied, which increases the complexity of feature interpretation. In addition, the dataset used in the study is insufficient, and the sample size is unbalanced. We selected abundant ASVs to construct networks, which will inevitably ignore the role of rare ASVs, which may play a more significant role in the pathogenesis of BV. This also is a limitation of our method. In the future, we will try to use different relevant measures to build the ASV correlation network, collect more data, and consider sample balance for research in order to obtain more reliable results, study the ecological mechanism of BV, and provide a theoretical basis for the diagnosis and treatment of BV.

## ACKNOWLEDGEMENTS

The authors thank Ravel et al. for their data support to this study.

### Funding

This study received funding from the scientific research fund of Yunnan Provincial Department of Education (No: 2021J0960). The funders had no role in study design, data collection and analysis, decision to publish, or preparation of the manuscript.

### Grant Disclosures

The following grant information was disclosed by the authors:
Yunnan Provincial Department of Education: 2021J0960.

### Competing Interests

The authors declare that they have no competing interests.

### Author Contributions

- Jie Li conceived and designed the experiments, performed the experiments, analyzed the data, prepared figures and/or tables, authored or reviewed drafts of the article, and approved the final draft.
- Yaotang Li conceived and designed the experiments, analyzed the data, authored or reviewed drafts of the article, and approved the final draft.

### Data Availability

The data and the code are available in the Supplemental Files.

### Supplemental Information

Supplemental information for this article can be found online at http://dx.doi.org/10.7717/peerj.14667#supplemental-information.

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
