# Peer review of "Detect feature edges for diagnosis of bacterial vaginosis"

_PeerJ, doi:10.7717/peerj.14667_

## Round 0.1 · original submission · Major Revisions

Before we can make a decision regarding the suitability of this manuscript for publication in PeerJ, a number of major issues need to be addressed in a revised manuscript. The issues are detailed in the reviewer comments, and all reviewer comments should be addressed in your revision. In particular, please carefully review references to the literature, including paraphrased comments which, as pointed at by reviewer 1, are not always an accurate representation of the original publication. Reviewer 1 also points out significant deficiencies in the detail provided describing the analytical methods reported in the manuscript. The methods employed should support the complete replication of your analysis by interested readers.

As indicated by reviewer 2, the datasets utilized for this study were generated over ten years ago and were the result of analytical methods and databases that are now considered out-of-date. To use these datasets to support the new analytical methods described in this study, you will need to go back to the original sequence files (available in SRA) and reanalyze those datasets utilizing more recent methods and a more up-to-date 16S database (such as SILVA).

Reviewer 1 ·

Basic reporting

There are mistakes in basic reporting. Lines 51-53 appear to slightly modify a sentence in Baker et al. (2014) which reads "Beck & Foster applied genetic programming, random forests, and logistic regression machine learning techniques on two BV datasets from Srinivasan et al. and Ravel et al. to hopefully discover BV related microbial relationships." The authors seem to have slightly modified this quote, mixing up the study in question. Other statements concerning published studies also seem like bad interpretations (for example line 42, the studies cited don't claim to "track the history of BV or to identify the pathogens of BV". They are generally looking at microbial associations with BV).

Experimental design

There is not enough information given about the methods used to allow for replication. Specifically, more information is needed about how the networks were constructed. Are the "networks" just pairwise correlations between microbial taxa?

The authors don't sufficiently justify their network approach, since a pairwise interaction is likely not providing any additional information than the abundances of the taxa individually. The top ranked "edges" identified in the study generally contain taxa associated with BV using individual correlations. What did the pairwise analysis add? This research doesn't appear to fill "an identified knowledge gap".

Validity of the findings

Impact and novelty aside, the study doesn't really provide meaningful replication.

The authors' general approach of using microbial network characteristics as the basis for improved BV diagnosis is potentially interesting but the included results seem insufficiently complete. The authors may consider network characteristics that aren't just pairwise correlations.

Additional comments

Some information given in the paper seems unnecessary. For example lines 128-136 include basic definitions.

Tables 1 and 2 contain no actual data and seem unnecessary. Other tables seem to have "h" characters replaced by "3"s. Maybe this is a file conversion artifact?

Inclusion of supplemental code is nice, although poorly documented. I did not evaluate the code's performance.

It seems unnecessary to include the raw data, since it has been previously published in the study it was obtained from.

Reviewer 2 ·

Basic reporting

For the most part the paper is well written, but there is a minor typo in the tables in which the letter "H" has been replaced with the number "3" in several species names.

Experimental design

My major concern with the paper is the use of the Ravel dataset from 2013, which was an OTU based method. This method was the gold standard at the time but has since been replaced by ASV (Amplicon Sequence Variant) based methods. This is important to mention because several key BV associated bacterium, and Lactobacillus species associated with healthy vaginal flora were difficult to detect and sometimes mislabeled using the OTU method. I would recommend rerunning this analysis using an ASV method to capture any potential mislabeling issues which may change the training set and then ultimately influence the resulting analysis.

I would also recommend adding additional information to the results section to clarify the significance of the results in each table. As written, I don't understand what the takeaway message from these tables is. I would also recommend adding more description to the table labels so that results are more accessible to a reader without a strong statistical background.

Validity of the findings

As stated above, I believe that the OTU based training data issue needs to be addressed before the validity of the results can be confirmed.

---

## Round 0.2 · Minor Revisions

Based on the reviews of your revised manuscript, a number of issues remain that need to be addressed before it can be accepted for publication. Please see the reviews below for details. But in general, the methods used for the analysis need to be clarified and better explained. In addition, the conclusions of the study need to be more clearly stated and explained.

Reviewer 1 ·

Basic reporting

No comment

Experimental design

The authors have improved reporting of their underlying methodology. The exception to this is the newly added ASV analysis. For example, in lines 95-100, what were the commands actually called? Line 95 indicates they were performed in Mothur, but the "vsearch" command doesn't show up in the Mothur manual. Both "usearch" and "vsearch" are listed, but it isn't clear why both would be used.

The two key paragraphs in the methods starting at lines 159 and 174 are difficult for me to understand. In the paragraph starting at line 159, each sample feature vector was left out and the remaining samples were used as the training set. How does this lead to an importance measure for individual network edges? Why was the network edge importance averaged over the 25 cross validation replicates rather than the subset of the 25 cross validation replicates that were contained in the comparison of interest? For example, the network edges important to the SBV vs HEA comparison should presumably only be averaged across the validation replicates containing either SBV or HEA designations.

In the paragraph starting at line 174, why were only 20 cross validation sets used? Weren't there a total of 25 feature vectors? The accuracy of each classifier is also stated as an average of the 20 cross validation sets, however, each comparison may have fewer than 20 relevant samples. For example, the SBV vs HEA comparison will presumably only include the SBV and HEA network feature vectors.

I must be missing something, and I'm not sure if it is my poor reading comprehension or a lack of clarity in the manuscript.

Validity of the findings

If the point of the study was to identify sub-networks of microbial taxa that correlate with BV, the authors state this was unsuccessful (line 279). This could be an interesting negative result, but I'm not convinced from the data they present. Is the only evidence for this the lack of consistent network edges identified as strong predictors of BV? Inconsistent high ranked edges could result from high levels of correlation between microbes leading to many redundant edges.

Additional comments

The abstract wording is difficult to parse. The list of conclusions separated by semi-colons might be better presented as an actual list.

Additionally, there are several typos.
Line 23: "a strengthen sub-network"
Lines 101,103: "EZbicloud" is probably "EzBioCloud"
Line 138: TP, TN, FP, and FN are defined but not P and N.
Line 242: something is missing, surprised feature selection method? supervised maybe?

Reviewer 2 ·

Basic reporting

no comment

Experimental design

On line 222 the author describes summing the ASV counts across the entire timepoint study. This should be fine, but I worry there is a potential to bias as people who become positive for BV will most likely undergo a shift in their microbiome which is a significant change but may not be a large fold shift and could be lost when only looking a the Top X. I would like the author to elaborate on this process some. Why was the Top 60 chosen as a cutoff? What percentage of the total ASVs present is captured in the Top 60?

Validity of the findings

no comment

---

## Round 0.3 · accepted · Accept

Thank you for responding to the reviewer critiques so quickly. I am happy to accept your manuscript for publication.